# Diagnostic Workflow in Competitive Athletes with Ventricular Arrhythmias and Suspected Concealed Cardiomyopathies

**DOI:** 10.3390/medicina57020182

**Published:** 2021-02-20

**Authors:** Maria Lucia Narducci, Michela Cammarano, Valeria Novelli, Antonio Bisignani, Chiara Pavone, Francesco Perna, Gianluigi Bencardino, Gaetano Pinnacchio, Massimiliano Bianco, Paolo Zeppilli, Vincenzo Palmieri, Gemma Pelargonio

**Affiliations:** 1Dipartimento di Scienze Cardiovascolari, Fondazione Policlinico Universitario Agostino Gemelli IRCCS, 00168 Rome, Italy; francesco.perna@policlinicogemelli.it (F.P.); gianluigi.bencardino@policlinicogemelli.it (G.B.); gaetano.pinnacchio@hotmail.it (G.P.); gemma.pelargonio@policlinicogemelli.it (G.P.); 2Unità Operativa Complessa di Medicina dello Sport e Rieducazione Funzionale, Fondazione Policlinico Universitario Agostino Gemelli IRCCS, Università Cattolica del Sacro Cuore, 00168 Rome, Italy; michela.cammarano01@gmail.com (M.C.); massimiliano.bianco@policlinicogemelli.it (M.B.); paolo.zeppilli@unicatt.it (P.Z.); vincenzo.palmieri@unicatt.it (V.P.); 3Fondazione Policlinico Universitario A. Gemelli IRCCS, UOC Genetica Medica, 00168 Rome, Italy; valeria.novelli@unicatt.it; 4Istituto di Medicina Genomica, Università del Sacro Cuore, L.go F. Vito 1, 00168 Rome, Italy; 5Istituto di Cardiologia, Università Cattolica del Sacro Cuore, L.go F. Vito 1, 00168 Rome, Italy; abisignani@hotmail.it (A.B.); chi.pavone@gmail.com (C.P.)

**Keywords:** athletes, ventricular arrhythmias, diagnostic workup, concealed cardiomyopathies

## Abstract

The diagnosis of structural heart disease in athletes with ventricular arrhythmias (VAs) and an apparently normal heart can be very challenging. Several pieces of evidence demonstrate the importance of an extensive diagnostic work-up in apparently healthy young patients for the characterization of concealed cardiomyopathies. This study shows the various diagnostic levels and tools to help identify which athletes need deeper investigation in order to unmask possible underlying heart disease.

## 1. Introduction

Sudden cardiac death (SCD) is an unexpected, devastating event, especially when occurring in young athletes, who are generally considered healthy individuals. 

Recent studies [1,2] have reported SCD during sporting activity as a small but significant fraction, 5%, of all SCD. It is estimated that sport-related SCD accounts for 15,000 deaths per year in North America and Europe [3].

According to comparative data from Italian studies, the risk of SCD in athletes is higher (2.3 deaths per 100,000 each year) with respect to non-athletes (0.9 per 100,000 each year). This risk is also strongly related to underlying cardiovascular diseases (2.1 per 100,000 each year from cardiovascular diseases [4]).

Up to 42% of athletes who suffer from SCD have an apparently normal heart and a negative toxicology screening [5]. Exercise is a recognized trigger for ventricular arrhythmias (VA) in athletes that harbor hereditary or congenital cardiac abnormalities, exposing them to increased risk of SCD [6]. An extensive diagnostic workflow could be very useful in order to identify concealed cardiomyopathies in young athletes (Figure 1). 

According to recent European Scociety of Cardiology (ESC) guidelines, screening strategies must be tailored to the target population and the specific disorders with the highest risk [7]. Special attention should be paid to ventricular arrhythmias (VA) as the initial clinical manifestation, elicited by vigorous exercise, of concealed cardiomyopathies, such as myocarditis, arrhythmogenic ventricular cardiomyopathy, and hypertrophic cardiomyopathy. 

## 2. First Level Cardiological Tools (Baseline ECG, Holter ECG, Stress Test, Transthoracic Echocardiography)

In Italy, the pre-participation clinical screening for competitive professional and non-professional athletes is established by Italian law and regional regulations [8]. The first-line evaluation includes medical history, physical examination, visual acuity test with the Snellen chart, resting 12-lead ECG, ECG stress testing, spirometry, and urine dipstick. 

An algorithm for the clinical management of athletes with premature ventricular beats (PVBs) [9] is provided. Its aim is to guide sports physicians towards a more in-depth diagnostic work-up in order to rule out an underlying pathological myocardial substrate. According to current guidelines, abnormal findings have implications for the athlete’s eligibility to take part in competitive sports activities [10].

The **ECG** represents an essential tool in the evaluation of athletes with PVBs. The most relevant ECG findings include repolarization abnormalities, such as T-wave inversion and ST-segment depression, pathological Q-waves, intraventricular conduction delays, ventricular pre-excitation, and a long QT interval. These ECG findings are classified as abnormal by current recommendations. Training-unrelated ECG findings warrant additional investigations to exclude myocardial disease [11]. The association of atypical ECG findings, such as left bundle branch block (LBBB) with an epsilon wave (small positive deflection or notch between the end of the QRS complex and onset of the T-wave in leads V1–V3) in the non-ectopic beats with VA of different morphologies, is highly suggestive of arrhythmogenic cardiomyopathy. Ventricular arrhythmias morphology can provide information about the location and origin of the arrhythmia. It represents a mainstay for the prognosis, and it must urge physicians to carry out further investigations. The most common VA morphology in athletes is the left bundle branch block (LBBB) one, which suggests the origin of the VA in the right ventricular outflow tract. Other common presentations include fascicular morphology, suggesting the origin of the VA within the conduction system, and right bundle branch block (RBBB), which derives from the left ventricle [12]. 

Following ECG examination, physicians must carry out an **exercise stress test**. This exam is useful for the assessment of PVBs associated with increasing workloads [13] and other abnormal findings suggestive of underlying cardiac disease, such as ST segment changes, abnormal blood pressure response to exercise, and impaired exercise tolerance. It is helpful in evaluating PVBs number and coupling intervals, as well as their morphology and pleomorphism.

Exercise-induced PVBs raise clinical suspicion because of their association with cardiomyopathy, myocarditis, and ion channel diseases, all of which are normally worsened by adrenergic stimulation [4,14,15,16]. On the other hand, PVBs that become less frequent or disappear with increasing exercise load are usually idiopathic and benign, with most of them being of infundibular origin (RV or LV outflow tract) [17,18]. Another important criterion is the complexity of the ventricular forms. Exercise-induced PVBs with multiple QRS morphologies, in particular beat-to-beat alternating morphologies (the so-called bi-directional pattern), are associated with a higher risk of effort-related SCD [9]. This arrhythmic feature may be an expression of catecholaminergic polymorphic ventricular tachycardia (VT), an inherited ion channel disease that predisposes to adrenergic-dependent VA, which can degenerate into ventricular fibrillation [19].

A **24 h ambulatory ECG monitoring** must be obtained in order to study the frequency and morphology of ectopic beats during the day. This exam is able to evaluate the so-called “arrhythmic burden”, namely the number of PVBs in 24 h and the presence of complex arrhythmias (pairs, triplets, or VT runs). It was demonstrated that athletes with frequent PVBs or non-sustained VT have an underlying cardiovascular disease more frequently than athletes with rare and isolated PVBs [20]. However, the presence of frequent PVBs is not necessarily synonymous with malignancy. As a matter of fact, benign extrasystole foci (typically located in the outflow tract of the right and, more rarely, left ventricle) can determine a very large number of PVBs at the 24 h Holter monitoring (more than 10,000/day), even in the absence of pathological substrate [21,22].

There is scientific evidence that athletes have ventricular arrhythmic burdens similar to those of the general population [14,23] and that prognosis is affected by the presence of an underlying substrate, rather than the number of PVBs. Accordingly, careful evaluation of familiar forms of SCD, such as cardiomyopathies or channelopathies, should be part of the diagnostic workup. Red flag symptoms include excessive bradycardia and lipotimic or syncopal episodes. 

**Echocardiography** is fundamental for the investigation of structural heart disease. With this exam, we can evaluate the heart’s global and regional function, both in systole and diastole, as well as wall motion abnormalities suggesting ischemic heart disease. It is also used to assess chamber size, valvular defects, and the presence of structural abnormalities suggestive of cardiomyopathy.

Echocardiography is the basal screening modality for the characterization of congenital coronary artery anomalies, which are one of the leading causes of ischemia-induced VA and SCD in the athlete. Zeppilli et al. [24] demonstrated that echocardiography can identify an aberrant coronary artery origin from the aorta in the vast majority of young athletes. With this examination, it was possible to manage and prevent cases of SCD in athletes who were sometimes completely asymptomatic [25]. 

However, echocardiography shows significant limitations in the evaluation of athletes with PVBs. It is unable to detect some conditions potentially associated with SCD during sports, with an apparently normal heart as the first presentation. Consequently, second-level diagnostic tools are needed for further evaluation.

## 3. Second-Level Diagnostic Tools (Cardiac MRI, Electroanatomic Mapping, Endomyocardial Biopsy)

**Cardiovascular magnetic resonance imaging** (MRI) can be of great value in the differential diagnosis between various cardiomyopathies, including hypertrophic cardiomyopathy (HCM), arrhythmogenic right ventricular cardiomyopathy (ARVC), left ventricle noncompaction cardiomyopathy (LVNC), and athlete’s heart [26]. This imaging tool can provide reliable and reproducible anatomical, functional, and tissue characterization information [27].

Cardiac MRI provides three-dimensional tomographic scans with high spatial and temporal resolution, allowing a clear delineation of endocardial and epicardial borders. Cardiac MRI can inspect the entire LV for abnormalities, including global/focal hypertrophy and wall motion disturbances. Late gadolinium-enhanced images (LGE) identify replacement myocardial fibrosis. Finally, the evaluation of pre-contrast (native) T1, post-contrast T1 mapping, and extracellular volume images (ECV) can identify diffuse myocardial edema. 

In particular, cardiac MRI is not useful in the definite diagnosis of early-stage cardiomyopathies, and some cardiac diseases, such as hypertrophic cardiomyopathy (HCM), arrhythmogenic right ventricular cardiomyopathy (ARVC), and initial forms of myocarditis with arrhythmic phenotype, can be misdiagnosed as athlete’s heart [23].

In a recent paper, we found that cardiac non-invasive imaging showed late gadolinium enhancement in only 30% of patients, with no difference between athletes and non-athletes [28]. Further evaluation with three-dimensional electroanatomic mapping (3D-EAM) as a second-level invasive approach, allowed us to identify abnormal low voltage areas in 60% of our population (mainly localized in the right ventricle), with no difference between athletes and non-athletes. 

**Electroanatomic mapping of ventricular substrate** may help to identify concealed myocardial diseases in competitive athletes presenting with recent-onset VA and an apparently normal heart [28,29]. It has the advantage of showing areas of abnormal voltages, which correspond to regions of diseased myocardium affected by scar tissue, fibrosis, or severe inflammation. Moreover, the study of electroanatomic substrate could increase the detection rate of earlier disease stages by guiding endomyocardial biopsy (EMB) towards areas of abnormal voltage.

3D-EAM-guided EMB allowed histopathological diagnosis in 50% of athletes showing abnormal low voltage areas [28]. Specifically, among 18 competitive athletes, our extensive diagnostic workup found 2 myocarditis, 1 myocardial focal replacement fibrosis, and 1 arrhythmogenic right ventricular cardiomyopathy. In a selected series of 13 athletes with a history of VA, Dello Russo et al. [29] reported no heart structure abnormalities at non-invasive imaging, but 3D-EAM-guided EMB allowed for the histological diagnosis of concealed cardiomyopathies in the whole population (seven myocarditis, five arrhythmogenic right ventricular cardiomyopathies, one contraction band myocardial necrosis). 3D-EAM-guided EMB was demonstrated to be safe and effective. Myocarditis is considered to be the main acquired cause of SCD in athletes and, as previously reported in the literature, high-intensity training may exacerbate myocarditis-related heart damage [4,30]. Analyzing in detail the different diagnostic tools in our study, 3D-EAM was found to be more reliable in detecting subtle cardiac arrhythmogenic substrates compared to cMRI [31]. 

## 4. Third-Level Diagnostic Tools (Genetic Screening)

In recent years, the advent of the Next Generation Sequencing (NGS) has made genetic testing in cardiology a useful diagnostic tool for detecting patients at high risk of SCD. 

Different sequencing panels have been developed to screen the main genes responsible for inherited cardiac conditions (ICC). In particular, more than 60 genes, including sarcomere and sarcomere-related genes, have been associated with inherited cardiomyopathies, but only a few of them are considered clinically actionable [32,33] and recommended in the screening.

It is important to highlight that, due to the limit in determining the clinical relevance of a variant and the variable yield of screening depending on the phenotype, genetic testing is highly recommended only in patients with an established clinical diagnosis or a strong suspicion of familial cardiomyopathy.

This issue becomes even more complicated in professional athletes, in whom cardiac remodeling can be a physiological consequence of sports activity. In these cases, it is crucial to carefully assess all the clinical steps, as previously stated, and to undertake a pre-test genetic counseling, in order to collect the family history of the proband and to explain the limitations of the test. 

The identification of a pathogenic (P) or likely pathogenic (LP) variant in “definitive” genes can provide several benefits to the patients’ management by (1) improving diagnostic accuracy, (2) identifying the molecular etiology of disease, (3) improving the prognosis, and (4) supporting therapeutic choices. Furthermore, cascade screening is always recommended, especially in asymptomatic young patients, to select family members at high risk [34].

## 5. Conclusions 

Extensive diagnostic work-up, including first-, second-, and third-level diagnostic tools, could be very helpful in the diagnosis of cardiomyopathies in apparently healthy young athletes with ventricular arrhythmic phenotype, no structural cardiac abnormalities, and normal resting ECG. We suggest this extensive diagnostic work-up including CMR, 3D-EAM mapping, 3D-EAM-guided EMB, and genetic testing in the characterization of the arrhythmogenic substrate in young athletes with complex VA.

## Figures and Tables

**Figure 1 medicina-57-00182-f001:**
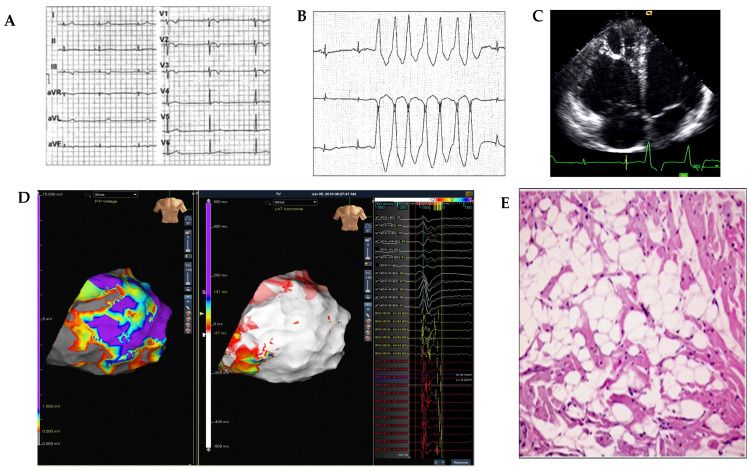
An example of diagnostic workup in a 21-year-old marathon runner presenting with palpitations and non-sustained ventricular tachycardia. Panel (**A**): sinus bradicardia on 12-lead-ECG; Panel (**B**): non-sustained ventricular tachycardia on 24 h-Holter monitor recordings; Panel (**C**): right ventricular re-enlargement with hyper-reflective moderator band by transthoracic echocardiography; Panel (**D**): dense scar area (grey area) in right free lateral wall detected by epicardial bipolar 3D-electroanatomic mapping and late potentials (red area) in the right ventricular scar area detected by activation map during sinus rhythm; Panel (**E**): histological diagnosis of arrhythmogenic right ventricular cardiomyopathy by endomyocardial biopsy.

## Data Availability

Not applicable.

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
