# Peer review of "Diagnostic Workflow in Competitive Athletes with Ventricular Arrhythmias and Suspected Concealed Cardiomyopathies"

_medicina, 2021, doi:10.3390/medicina57020182_

Round 1
Reviewer 1 Report
In this study, Narducci et al. listed first to third-level of analytic cardiological tools that could improve the diagnosis of concealed cardiomyopathies in athletes with ventricular arrhythmic phenotype but otherwise normal-appearing hearths. The authors mentioned how suspicious findings in first-line evaluation can be supported by deeper investigations from medical imaging to genetic screening. They elaborate on how this level scheme of diagnostics helps to detect and differentiate idiopathic irregularities from those with underlying cardiomyopathies that could lead to sudden cardiac death.
The review is an informative and well-structured overview of the diagnostic possibilities. However, the conclusion lacks suggestions for adapted diagnostic guidelines including these deeper investigations, and an outlook on how the methods can decrease the cardiac risk of healthy appearing athletes is missing. Overall, it's a positive new insight into the topic.
Author Response
We thank the Reviewer for reading our manuscript carefully and for providing detailed comments and suggestions that were helpful in improving the manuscript.
Particularly. according to the Reviewer, we, changed the conclusion including suggestions for adapted diagnostic guidelines in the revised manuscript. We add the following sentence in Conclusion session:
"We suggest this extensive diagnostic work-up including CMR, 3DEAM mapping, 3D-EAM-guided EMB and genetic testing in the characterization of the arrhythmogenic substrate in young athletes with complex VA".

Reviewer 2 Report
This manuscript is written well, however this need to be revised several points.
1) Introduction
Could you describe this more exactly? Because I did not understand this manuscript's purpose in this section.
2) Methods
Could you show that methods of the selection for the references?
Because, it was not show how to select of the references in this section.
Author Response
We thank the Reviewer for his/her careful reading and for his/her important suggestions. We have addressed all of the Reviewer’s comments.
1) According to the Reviewer, we revised the Introduction session specifying the different screening strategies in athletes with complex ventricular arrhythmias and concelead cardiomyopathies. The purpose of our review is to evaluate all literature about extensive diagnostic work-up in young athletes with complex VAs including non-invasive imaging (echocardiography and cardiac magnetic resonance ), and invasive three dimensional-electroanatomical mapping 3D.EAM), followed by endomyocardial biopsy (EMB) and genetic testing when clinically indicated.
Consequently , we added the following sentence in the Introduction Session:
"According to recent ESC guidelines, screening strategies must be tailored to the target population and the specific disorders with highest risk [7]. Special attention should be paid to ventricular arrhythmias (VA) as initial clinical manifestation, elicited by vigorous exercise, of concealed cardiomyopathies, as myocarditis, arrhythmogenic ventricular cardiomyopathy, hypertrophic cardiomyopathy"
2) With regard to the Methods, we have reviewed all articles published from 2013 to 2020 in the field of "athletes and ventricular arrhythmias", including non invasive and invasive diagnostic tools.
